# Information Theoretic Lower Bounds for Information Theoretic Upper Bounds

**Roi Livni**
Department Electrical Engineering
Tel Aviv University
`rlivni@tauex.tau.ac.il`

## Abstract

We examine the relationship between the mutual information between the output model and the empirical sample and the generalization of the algorithm in the context of stochastic convex optimization. Despite increasing interest in information-theoretic generalization bounds, it is uncertain if these bounds can provide insight into the exceptional performance of various learning algorithms. Our study of stochastic convex optimization reveals that, for true risk minimization, dimension-dependent mutual information is necessary. This indicates that existing information-theoretic generalization bounds fall short in capturing the generalization capabilities of algorithms like SGD and regularized ERM, which have dimension-independent sample complexity.

## 1 Introduction

One of the crucial challenges facing contemporary generalization theory is to understand and explain the behavior of overparameterized models. These models have a large number of parameters compared to the available training examples. But nonetheless, they tend to perform well on unseen test data. The significance of this issue has become more pronounced in recent years, as it has become evident that many state-of-the-art learning algorithms are highly overparameterized Neyshabur et al. [2014], Zhang et al. [2021]. The classical generalization bounds, which are well designed to describe the learning behavior of underparameterized models, seem to fail to explain these algorithms.

Understanding the success of overparameterized models seems challenging. Partly, due to the counter-intuitive nature of the process. Common wisdom suggests that, inorder to learn, one has to have certain good bias of the problem at hand, and that in learning we need to restrict ourselves to a class of models that cannot overfit the data. This intuition has been justified by classical learning models such as PAC learning Valiant [1984] as well as regression Alon et al. [1997]. In these classical models, it can be even demonstrated Vapnik and Chervonenkis [2015], Blumer et al. [1989] that learning requires more examples than the capacity of the class of model to be learnt, and that avoiding interpolation is necessary for generalization. These results, though, are obtained in distribution-independent setups where one assumes worst-cast distributions over the data.

For this reason, researchers have been searching for new, refined models, as well as improved generalization bounds that incorporate distributional as well as algorithmic assumptions. A promising approach, in this direction, tries to connect the generalization performance to the amount of information the learner holds regarding the data Russo and Zou [2019], Xu and Raginsky [2017], Bassily et al. [2018]. For example, Xu and Raginsky [2017] demonstrated an upper bound on the generalization gap which informally states:

$$\text{generalization gap}(w_S) = O\left(\sqrt{\frac{I(w_S, S)}{|S|}}\right) \tag{1}$$

Namely, given an empirical sample $S$, and an output model $w_S$, the difference between its empirical error and its true error can be upper bounded by $I(w_S, S)$, the mutual information between these two random variables. Notice that Eq. (1) does not depend on any trait of the class of feasible models to be considered. In particular, it does not depend, apriori, on number of "free parameters", or a complexity measure such as VC dimension, the dimension of $w_S$, and not even on some prior distribution. However, it remains a question whether this method and technique can be useful in analysing state-of-the-art learning algorithms. While there has been a lot of work trying to establish the success of learning algorithms in various setups Neu et al. [2021], Xu and Raginsky [2017], Bu et al. [2020], Aminian et al. [2021], Pensia et al. [2018], many of the established bounds are opaque, and often there is no comprehensive end-to-end analysis that effectively illustrates how generalization is to be bounded by Eq. (1) and *simultaneously* obtain good empirical performance. In fact, there is also evidence Carlini et al. [2021], Feldman [2020] that memorizing data is required, in some regimes, for effective learning. Towards better understanding, we will focus, in this work, on the setup of *Stochastic Convex Optimization* Shalev-Shwartz et al. [2009] (SCO), and provide accompanying lower bounds to Eq. (1) that will describe how much mutual information is *necessary* for learning.

**SCO as a case study for overparametrization:**  SCO is a very clean and simple setup where a learner observes noisy instances of (Lipschitz) convex functions, defined in $\mathbb{R}^d$, and is required to minimize their expectation. On the one hand, it provides simple, amenable to rigorous analysis, definitions of learnability and learning. On the other hand, this model is the cradle of prototypical algorithms such as Gradient Descent (GD) and Stochastic Gradient Descent (SGD), as well as accelerated methods, which are the workhorse behind state-of-the-art optimization methods.

Moreover, SCO is an ideal model for understanding overparameterization. It is known Feldman [2016] that in this setup, $\Omega(d)$ examples are needed in order to avoid overfitting. In fact, even concrete algorithms such as GD and regularized-GD may overfit unless they observe dimension-dependent sample size Amir et al. [2021a,b]. In other words, the capacity of the model and its ability to overfit does indeed scale with the dimension. Nevertheless, it is also known that *some* algorithms do learn with far fewer examples. For example SGDHazan et al. [2016], Regularized-ERMShalev-Shwartz et al. [2009], Bousquet and Elisseeff [2002] and a stable variant of GD Bassily et al. [2020] all learn with $O(1/\varepsilon^2)$ examples, a dimension independent magnitude. To put it differently, learning in SCO is not just a question of finding the empirical risk minimizer, but also a question of *how* – what algorithm was used, and learning is not demonstrated by naive uniform convergence bounds that scale with the number of parameters in the model.

Therefore, SCO is a natural candidate to study how information theoretic bounds play a role in learning. We might even hope that these bounds shed light on why some algorithms succeed to learn while others fail. Existing algorithms don't avoid memorizing the data, but it is unclear if holding information on the data is *necessary*. So we start here with the simplest question:

> What is the smallest amount of mutual information required for learning in SCO?

Our main result shows that, in contrast with the dimension-independent learnability results in this setup, the information between the model and the sample has to be *dimension-dependent*. As such, the complexity of the class appears implicitly in Eq. (1). As a result, carrying $\Omega(d)$ bits of information over the sample is *necessary* for learning at optimal rates, and Eq. (1) doesn't yield the optimal generalization performance of algorithms such as Regularized ERM, SGD and stable-GD.

## 1.1 Related Work

Information-theoretic generalization bounds have a long history of study in ML theory McAllester [1998, 1999], Langford and Shawe-Taylor [2002]. Generalization bounds that directly relate to the information between output and input of the learner initiated in the works of Xu and Raginsky [2017], Bassily et al. [2018], Russo and Zou [2019], Bassily et al. [2018] demonstrated limitations for such generalization bounds, for proper ERM learners, and Livni and Moran [2020] showed that any learner (proper or not), that learns the class of thresholds must leak unbounded amount of information. In

this work we focus on stochastic optimization and on learning Lipschitz functions. In the setup of SCO the above is not true, and one can construct learners that leak $\tilde{O}(d)$ bits of information (see Proposition 1). But we would like to know whether information-theoretic bounds behave like the uniform convergence bounds (dimension dependent) or capture the minmax learning rates (dimension independent).

Several lines of works applied the analysis of Xu and Raginsky [2017] to provide algorithmic-dependent analysis in the context of stochastic optimization. Pensia et al. [2018] and followup improvements Rodríguez-Gálvez et al. [2021], Negrea et al. [2019], Haghifam et al. [2020] provided information-theoretic generalization bounds for Stochastic Gradient Langevine Dynamics (SGLD) and Neu et al. [2021] extends the idea to analyze vanilla SGD. Aminian et al. [2021] also provides full characterization of the closely related, Gibbs Algorithm and shows that the information can inverseley scale with the sample bounds. The bounds in these works are *implicitly* dimension independent which may seem contradictory to the result established here. Importantly, the above bounds may depend on hyper-parameters such as noise in SGLD and temparature in Gibbs algorithm, and these hyperparameters may affect the optimization performance of the algorithms. The bounds we obtain are applicable only to algorithms with non-trivial true risk, which depends on hyperparameter choice, and as such there is no real contradiction. Taken together, the noise, for example, in SGLD needs to scale with the dimension in order to obtain non-trivial information-theoretic bounds, but that will lead to a large empirical error. Similarly, if the temperature in the Gibbs algorithm doesn't scale with the dimension, one can potentially achieve small empirical error but at the expanse of high information.

Most similar to our work, recently, Haghifam et al. [2022] provided the first set of limitations to information theoretic generalization bounds. They focus on the Gradient Descent method and perturbed variants of it, and provide limitations to both MI bounds as well as conditional mutual information (CMI) bounds Steinke and Ullman [2015] and their individual sample variants Bu et al. [2020], Negrea et al. [2019], Haghifam et al. [2020], Zhou et al. [2022]. In contrast, we focus on the mutual information bound (as well as its individual sample version of Bu et al. [2020]), but we provide bounds that are irrespective of the algorithm.

The key idea behind our lower bound proof builds on privacy attacks developed in the differntial privacy literature Bun et al. [2014], Kamath et al. [2019], Steinke and Ullman [2015]. In the context of classification, lower and upper bounds techniques Alon et al. [1997], Bun et al. [2020] were successfully imported to obtain analogous information-theoretic bounds Livni and Moran [2020], Pradeep et al. [2022]. In optimization, though, bounds behave slightly different and therefore, bounds from classification cannot be directly imported to the context of optimization.

## 2 Setup and Main Results

We begin by describing the classical setup of Stochastic Convex Optimization (SCO), following Shalev-Shwartz et al. [2009]. In this model, we assume a domain $\mathcal{Z}$, a parameter space $\mathcal{W} \subseteq \mathbb{R}^d$ and a function $f(w, z)$, termed *loss* function. The function $f$ satisfies that for every $z_0 \in \mathcal{Z}$, the function $f(w, z_0)$ as a function over the parameter $w$ is convex and $L$-Lipschitz.

For concreteness, we treat $L$ as a constant, $L = O(1)$, and we concentrate on the case where $\mathcal{W}$ is the unit ball. Namely:
$$\mathcal{W} = \{w : \|w\| \le 1\}.$$

As we mostly care about lower bounds, these won't affect the generality of our results. Given a distribution $D$, the expected loss of a parameter $w$ is given by
$$L_D(w) = \mathbb{E}_{z \sim D}[f(w, z)].$$

The excess true risk of $w$, with respect to distribution $D$, is denoted as:

$$\Delta_D(w) = L_D(w) - L_D(w^\star), \quad \text{where, } L_D(w^\star) := \min_{w \in \mathcal{W}} \mathbb{E}_{z \sim D}[f(w, z)].$$

We also denote the excess empirical risk, given sample $S = \{z_1, \ldots, z_m\}$:

$$\Delta_S(w) = \frac{1}{m} \sum_{i=1}^{m} f(w, z_i) - \min_{w \in \mathcal{W}} \frac{1}{m} \sum_{i=1}^{m} f(w, z_i).$$

**Leranbility** We will focus here on the setup of *learning in expectation*. In particular, a learning algorithm $A$ is defined as an algorithm that receives a sample $S = (z_1, \ldots, z_m)$ and outputs a parameter $w_S^A$. We will normally supress the dependence of the parameter $w_S^A$ in $A$ and simply write $w_S$. The algorithm $A$ is said to *learn* with sample complexity $m(\varepsilon)$ if it has the following property: For every $\varepsilon > 0$, if $S$ is a sample drawn i.i.d from some unknown distribution $D$, and $|S| \geq m(\varepsilon)$ then:

$$\mathbb{E}_{S \sim D^m} [\Delta_D(w_S)] \leq \varepsilon.$$

A closely related setup requires that the learner succeeds with high probability. Standard tools such as Markov's inequality and boosting the confidence Schapire [1990] demonstrate that the two definitions are essentially equivalent in our model.

**Information Theory** We next overview basic concepts in information theory as well as known generalization bounds that are obtained via such information-theoretic quantities. We will consider here the case of discrete random variables. We elaborate more on this at the end of this section, and how our results extend to algorithms with continuous output. Therefore, throughout, we assume a discrete space of possible outcomes $\Omega$ as well as a distribution $\mathbb{P}$ over $\Omega$. Recall that a random variable $X$ that takes values in $\mathcal{X}$ is said to be distributed according to $P$ if $\mathbb{P}(x = X) = P(x)$ for every $x \in \mathcal{X}$. Similarly two random variables $X$ and $Y$ that take values in $\mathcal{X}$ and $\mathcal{Y}$ respectively have joint distribution $P$ if

$$\mathbb{P}(x = X, y = Y) = P(x, y).$$

For a given $y \in \mathcal{Y}$, the conditional distribution $P_{X|y}$ is defined to be $P(x \mid y = Y) = \frac{P(x,y)}{\sum_x P(x,y)}$, and the marginal distribution $P_X : \mathcal{X} \to [0, 1]$ is given by $P_X(x) = \sum_y P(x, y)$. If $P_1$ and $P_2$ are two distributions defined on a discrete set $\mathcal{X}$ then the KL divergence is defined to be:

$$D_{KL}(P_1 \| P_2) = \sum_{x \in \mathcal{X}} P_1(x) \log \frac{P_1(x)}{P_2(x)}.$$

Given a joint distribution $P$ that takes values in $\mathcal{X} \times \mathcal{Y}$ the mutual information between random variable $X$ and $Y$ is given by

$$I(X; Y) = \mathbb{E}_Y \left[ D_{KL}(P_{X|Y} \| P_X) \right].$$

We now provide an exact statement of Eq. (1)

**Theorem** (Xu and Raginsky [2017]). *Suppose $f(w, z)$ is a bounded by 1 loss function. And let $A$ be an algorithm that given a sample $S = \{z_1, \ldots, z_m\}$ drawn i.i.d from a distribution $D$ outputs $w_S$. Then*

$$\mathbb{E}_S \left[ L_D(w_S) - \frac{1}{m} \sum_{i=1}^m f(w, z_i) \right] \leq \sqrt{\frac{2I(w_S, S)}{m}}. \tag{2}$$

**Remark on Continuous Algorithms** As stated, we focus here on the case of algorithms whose output is discrete, and we also assume that the sample is drawn from a discrete set. Regarding the sample, since we care about lower bounds and our constructions assume a discrete set, there is no loss of generality here. Regarding the algorithm's output, in the setup of SCO there is also no loss of generality in assuming the output is discrete. Indeed, we can show that if there exists a continuous algorithm with sample complexity $m_0(\varepsilon)$, and bounded mutual information over the sample, then there exists also a discrete algorithm with sample complexity $m(\varepsilon) = m_0(O(\varepsilon))$ with even less mutual information.

To see that notice that, since we care about minimizing a Lipschitz loss function, given any accuracy $\varepsilon$, we can take any finite $\varepsilon$-approximation subset of the unit ball and simply project our output to this set. Namely, given output $w_S$, where $S > m(\varepsilon)$, we output $\bar{w}_S$, the nearest neighbour in the $\varepsilon$-approximation sub set. Because $L_D$ is $O(1)$ Lipschitz, we have that

$$|L_D(\bar{w}_S^A) - L_D(w^\star)| \leq |L_D(\bar{w}_S^A) - L_D(w_S^A)| + |L_D(w_S^A) - L_D(w^\star)| \leq O(\varepsilon),$$

hence up to a constant factor the algorithm has the same learning guarantees. On the other hand, due to data processing inequality:

$$I(\bar{w}_S^A, S) \leq I(w_S^A, S).$$

## 2.1 Main Result

Eq. (2) provides a bound over the difference between the expected loss of the output parameter and the empirical loss. Without further constraints, it is not hard to construct an algorithm that carries little information on the sample. But, to obtain a bound over the excess risk of the parameter $w_S$, one is required not only to obtain a generalization error gap but also to non-trivially bound the empirical risk. The following result shows that requiring both has its limits:

**Theorem 1.** *For every $0 < \varepsilon < 1/54$ and algorithm A, with sample complexity $m(\varepsilon)$, there exists a distribution D over a space $\mathcal{Z}$, and loss function $f$, 1-Lipschitz and convex in w, such that, if $|S| \geq m(\varepsilon)$ then:*

$$I(w_S; S) \geq \sum_{i=1}^{m} I(w_S, z_i) = \tilde{\Omega}\left(\frac{d}{\varepsilon^5 \cdot m^6(\varepsilon)}\right). \tag{3}$$

Theorem 1 accompanies the upper bound provided in Eq. (2) and shows that, while the generalization gap is bounded by the mutual information, the mutual information inversely scales with the optimality of the true risk. Taken together, for any algorithm with non-trivial learning guarantees, there is at least one scenario where it must carry a dimension-dependent amount of information on the sample or require a large sample. Indeed, either $m(\varepsilon) = \Omega(\sqrt[6]{d/\varepsilon^5})$, (and then, trivially, the sample complexity of the algorithm scales with the dimension) or, via Eq. (3), we obtain dimension dependent mutual information, and in turn, Eq. (2) is non-vacuous only if the sample is larger than the dimension.

The first inequality is standard and follows from standard chain rule argument (see e.g. [Bu et al., 2020, proposition 2]). The second inequality lower bounds the information with the individual samples. Recently, Bu et al. [2020] obtained a refined bound that improves over Xu and Raginsky [2017] by bounding the generalization with the information between the individual samples and output. Theorem 1, together with subadditivity of the square function, shows that the individual sample bound of Bu et al. [2020] can also become dimension dependent.

## 3 Discussion

Our main result shows *a necessary* condition on the mutual information between the model and empirical sample. Theorem 1 shows that for any algorithm with non-trivial learning guarantees, there is at least one scenario where it must carry a dimension-dependent amount of information on the sample or require a large sample. A natural question, then, is whether natural structural assumptions may circumvent the lower bound and allow to still maintain meaningful information theoretic bounds in slightly different setups. To initiate a discussion on this we begin by looking deeper into the concrete construction at hand in Theorem 1. We notice (see Section 4 and the supplementary material) that the construction we provided for Theorem 1 relies on a distribution $D$ that is always supported on functions of the form:

$$f(w, z) = \|w - z\|^2 = \|w\|^2 - 2w \cdot z + 1, \quad z \in \{-1/\sqrt{d}, 1/\sqrt{d}\}^d. \tag{4}$$

The constant 1 has no effect over the optimization, nor on the optimal solution, therefore we can treat $f$ as equivalent to the following function

$$f \equiv \|w\|^2 - 2w \cdot z.$$

The distribution over the element $z$ is also quite straightforward and involves only bias sampling of the coordinates. The function $f$, then, is arguably the simplest non-linear convex function that one can think of and as we further discuss it holds most if not all of the niceties a function can hold that allow fast optimization – it is strongly convex, smooth, and in fact enjoys generalization bounds that can even be derived using standard uniform convergence tools. Indeed for any choice $w_S$

$$\mathop{\mathbb{E}}_{S \sim D^m} L_D(w_S) - \frac{1}{m}\sum_{i=1}^{m} f(w_S, z_i) = 2 \mathop{\mathbb{E}}_{S \sim D^m} \sup_{\|w\| \leq 1} [\frac{1}{m}\sum_{i=1}^{m} w \cdot z_i - \mathop{\mathbb{E}}_{z \sim D}[w \cdot z]]$$

$$\leq O(1/\sqrt{m})$$

Where the last inequality follows from a standard Rademacher bound over the complexity of linear classifiers (see Shalev-Shwartz and Ben-David [2014]). In other words, while under further structural

assumptions one might hope to obtain meaningful information theoretic bounds, it should be noted that such structural assumptions must exclude a highly simplistic class of functions that are in fact extremely *easy* to learn and even enjoy dimension independent uniform convergence bounds.

We now discuss further the niceties of this function class and the implications to refined algorithmic/distributional-dependent generalization bound via the information-theoretic bounds. We begin by analyzing algorithms that achieve the minimax rate.

**Algorithmic-dependent generalization bounds**   As discussed, it is known that SGD Hazan et al. [2016], regularized-ERM Shalev-Shwartz et al. [2009], Bousquet and Elisseeff [2002], as well as stabilized versions of Gradient Descent Bassily et al. [2020] have the following minmax rate for learning $O(1)$-Lipscthiz convex functions over an $O(1)$-bounded domain:

$$\mathbb{E}\left[\Delta_D(w_S)\right] = O(1/\sqrt{m}).$$

Plugging the above in Eq. (3) we obtain that any such algorithm *must* carry at least $\Omega(d/m^{5/2})$ bits of information. which entails an information theoretic bound in Eq. (2) of $O(\sqrt{d}/m^{7/4})$. This exceeds the true excess risk of such algorithms when $d \gg m$ and becomes a vacuous bound when we don't assume the sample scales with the dimension (even though the algorithm perfectly learns in this setup). Now, we do not need Theorem 1 to obtain generalization gaps over these concrete algorithms, as a more direct approach would do. But, one might hope that by analyzing noisy versions of these algorithms, or some other forms of information-regularization, we could obtain some insight on the generalization of these algorithms. But, our lower bound applies to any algorithm with meaningful information-theoretic generalization bound. In particular, adding noise, for example, either makes the algorithm diverge or the noise is too small to delete enough information.

**Distributional-dependent generalization bounds**   Next, we would like to discuss the implications to distributional assumptions. The above discussion shows that any trait of an algorithm that makes it optimal is not captured by the amount of information it holds on the data (in the setup of SCO). Distinct from SCO, in practice, many of the underlying problems can be cast into binary classification where it is known Blumer et al. [1989] that without further distributional assumptions learnability cannot be separated from uniform convergence as in SCO.

An interesting question, then, is if information theoretic generalization bounds can be used to obtain *distribution-dependent* bounds.

The function $f(w, z)$ in Eq. (4) is known to be *strongly-convex* for every $z$. Recall that a function $f$ is called 1-strongly convex if $f - \frac{1}{2}\|w\|^2$ is convex. It is known Shalev-Shwartz et al. [2009] that any ERM over a strongly convex will achieve suboptimality:

$$\mathbb{E}[\Delta_D(w_S)] = O(1/m).$$

Moreover, the above result can be even strengthened to any approximate empirical risk minimizer, that is a minimizer with an additive $\Theta(1/m^2)$ error over the empirical risk Shalev-Shwartz et al. [2009], Amir et al. [2021b]. But even further, for the particular structure of $f$, which is a regularized linear objective, by [Sridharan et al., 2008, Thm 1] we have:

$$\mathbb{E}[\Delta_D(w_S)] \leq \tilde{O}(\mathbb{E}[\Delta_S(w_S)] + 1/m).$$

Together with Theorem 1 we obtain the following algorithmic-independent result:

**Theorem 2.** *There exists a family of distributions $\mathcal{D}$, such that for any algorithm A and $m > 3$, there exists a distribution $D \in \mathcal{D}$ such that if $\Delta_S(w_S) \leq 1/54$ and $|S| > m$ then :*

$$I(w_S, S) = \tilde{\Omega}\left(\frac{d}{m^6 \cdot (\mathbb{E}[\Delta_S(w_S)] + 1)^5}\right),$$

*but for any algorithm A and distribution $D \in \mathcal{D}$:*

$$\mathbb{E}[\Delta_D(w_S)] = \tilde{O}\left(\mathbb{E}[\Delta_S(w_S)] + 1/m\right).$$

In other words, even without algorithmic assumptions, we can construct a class of distributions which make the problem *easy to learn*, but the information bounds are still dimension dependent. In

particular, for any algorithm such that $\Delta_S(w_S) = O(1/m)$ we will have that the bound in Eq. (2) is order of:

$$\mathbb{E}[\Delta_D(w_S)] = \tilde{O}\left(\sqrt{d/m}\right). \tag{5}$$

**Comparison to uniform convergence bounds:** Notice that Eq. (5) is the standard generalization bound that can be obtained for *any* ERM algorithm in the setting of stochastic convex optimization. In detail, through a standard covering number argument (Shalev-Shwartz et al., 2009, thm 5) it is known that, when $f$ is 1-Lipschitz (not necessarily convex even):

$$\sup_{w \in \mathcal{W}} \mathbb{E}_S \left[ L_D(w) - \frac{1}{m} \sum_{i=1}^{m} f(w, z_i) \right] \leq \tilde{O}(\sqrt{d/m}).$$

In other words, the bound we obtain in Eq. (5) can be obtained for *any* algorithm, with minimal assumptions, irrespective of the mutual information between output and sample. We remark, though, that, through a similar covering argument, one can show that indeed there are algorithms where one can recover the above bound via information-theoretic reasoning.

**Proposition 1.** *Given a* 1-*Lipschitz function* $f$*, there exists an algorithm A that given input sample S outputs a parameters* $w_S \in \mathcal{W}$*: such that*

$$\frac{1}{m} \sum_{i=1}^{m} f(w_S, z_i) \leq \min_{w^\star \in \mathcal{W}} \frac{1}{m} \sum_{i=1}^{m} f(w^\star, z_i) + \sqrt{\frac{d}{m}}, \tag{6}$$

*and*

$$I(w_S, S) = \tilde{O}(d \log m).$$

*In particular,*

$$L_D(w_S) - L_D(w^\star) = \tilde{O}(\sqrt{d/m}).$$

*Sketch.* The result follows a simple covering argument. In particular, it is known that there exists a finite subset $\bar{\mathcal{W}} \subseteq \mathcal{W}$ of size $|\bar{\mathcal{W}}| = O\left(\sqrt{m}^d\right)$ such that for every $w \in \mathcal{W}$ there is $\bar{w} \in \bar{\mathcal{W}}$ such that $\|w - \bar{w}\| \leq \sqrt{d/m}$ (e.g. Wu [2017]). Now. we consider an ERM that is restricted to the set $\bar{W}$. By Lipschitness we have that Eq. (6) holds. The information is bounded by the entropy of the algorithm and we have that

$$I(W_S, S) \leq H(W_S) \leq \log |\bar{\mathcal{W}}| = O(d \log m).$$

The genearlization gap can be bounded via Eq. (2) (or standard union bound). ∎

**CMI-bounds** Similarly to our setting, also in the setting of PAC learning, Livni and Moran [2020], Bassily et al. [2018] provided limitations to information theoretic generalization bounds. Specifically, they showed that such bounds become vacous in the task of learning thresholds, and the information between output and sample may be unbounded. To resolve this issue, which happens because of the numerical precision required by a thresholds learner, Steinke and Zakynthinou [2020] introduced generalization bounds that depend on *conditional mutual information* (CMI). They provided CMI bounds that can be derived from VC bounds, compression bounds etc... which in a nutshell means that they are powerful enough to achieve tight learning rates for VC classes such as thresholds. However, the issue in SCO is not comparable. As we show in Proposition 1 already the classical information-theoretic bounds are powerful enough to demonstrate generalization bounds that can be derived via union bound or uniform convergence. In PAC learning, these bounds are also tight, but in SCO such bounds are dimension-dependent. In that sense, there is no analog result to the limitation of learning thresholds. Partly because there is no need for infinite precision in SCO. In SCO, though, we require a separation from uniform convergence bounds, or even more strongly - from dimension dependent bounds. While Haghifam et al. [2020] does demonstrate certain algorithmic-dependent limitations for GD and perturbed GD algorithms, one might hope that, similar to PAC learning, here too CMI-bounds might be able to capture optimal dimension-independent rates for some algorithms.

More formally, given a distribution $D$ over $\mathcal{Z}$, we consider a process where we draw i.i.d two random samples $Z = S_1 \times S_2$, where $S_1 = \{z_1^0, \ldots, z^0 m\} \sim D^m$ and $S_2 = \{z_1^1, \ldots, z_m^1\} \sim D^m$. Then we define a sample $S$ by randomly picking $z_i = z_i^0$ w.p. 1/2 and $z_i = z_i^1$ w.p. 1/2 (independently from

$z_1, \ldots, z_{i-1}, z_{i+1}, \ldots, z_m$). Then, Steinke and Zakynthinou [2020] showed that, similarly to Eq. (17), the generalization of an algorithm can be bounded in terms of

$$\text{generalization}(A) = O\left(\sqrt{\text{CMI}_m(A)/m}\right), \quad \text{CMI}_m(A) = I\left(w_S, S \mid Z\right). \tag{7}$$

Recall that, given a random variable $Z$, the conditional mutual information between r.v. $X$ and r.v. $Y$ is defined as:

$$I\left(X; Y \mid Z\right) = \mathbb{E}_Z\left[\mathbb{E}_Y\left[D_{KL}\left(P_{X|Z} \| P_{X|Y,Z}\right)\right]\right].$$

One can show that $\text{CMI}_m(A) = O(m)$. The simplest way to achieve a dimension independent generalization bound would be to subsample. If algorithm $A$ observes a sample $S$ of size $m$ and subsamples $f(m)$ samples, then we trivially obtain the generalization gap: $O(1/\sqrt{f(m)})$. Taking Eq. (7) into consideration by subsampling $O(\sqrt{m})$ examples, Eq. (7) will lead to a dimension independent generalization bound of $O(1/\sqrt[4]{m})$. So it is possible to obtain dimension independent bounds through the CMI framework, nevertheless it is still not clear if we can beat the above, nonoptimal and naive subsampling bound

**Open Problem.** *Is there an algorithm A in the SCO setup that can achieve*

$$CMI_m(A) = \tilde{o}(\sqrt{m}),$$

*as well as*

$$\mathbb{E}\left[\Delta_D(A)\right] = O(1/\sqrt{m}).$$

## 4 Technical overview

We next outline the key technical tools and ideas that lead to the proof of Theorem 1. A full proof of Theorem 1 is provided in the supplementary material. As discussed we draw our idea from the privacy literature Steinke and Ullman [2015], Kamath et al. [2019], Bun et al. [2014] and build on fingerprinting Lemmas Boneh and Shaw [1998], Tardos [2008] to construct our "information attacks". We employ a simplified and well-tailored version of the fingerprinting Lemma, due to Kamath et al. [2019], to construct a lower bound on the correlation between the output and the data. From that point our proof differ from standard privacy attacks.

Given the above correlation bound, we now want to lower bound the mutual information between the two correlated random variables (i.e. the output of the algorithm and the empirical mean). Surprisingly, we could not find in the literature an existing lower bound on the mutual information between two correlated random variables. The next Lemma provides such a lower bound and as such may be of independent interest.. We next depict these two technical Lemmas that we need for our proof – the fingerprinting Lemma due to Kamath et al. [2019] and a lower bound on the mutual information between two correlated random variables.

For describing the fingerprinting Lemma, let us denote by $U[-1/3, 1/3]$ the uniform distribution over the interval $[-1/3, 1/3]$, and given $p \in [-1/3, 1/3]$, we denote by $Z_{1:m} \sim U_m(p)$ a random process where we draw $m$ i.i.d random variables $Z_1, \ldots, Z_m$ where $Z \in \{\pm 1\}$ and $\mathbb{E}[Z] = p$:

**Lemma 1** (Fingerprinting Lemma (Kamath et al. [2019])). *For every $f : \{\pm 1\}^m \to [-\frac{1}{3}, \frac{1}{3}]$, we have:*

$$\mathbb{E}_{P \sim U[-1/3,1/3]} \mathbb{E}_{Z_{1:m} \sim U_m(P)} \left[\frac{1 - 9P^2}{9 - 9P^2} \cdot (\hat{f}(Z_{1:m}) - P) \cdot \sum_{i=1}^{m} (Z_i - P) + (\hat{f}(Z_{1:m}) - P)^2\right] \geq \frac{1}{27}$$

The above theorem shows that if a random variable $\hat{f}(Z_{1:m})$ uses the sample to non-trivially estimate the random variable $P$, then the output must correlate with the empirical mean. Notice that, in particular, it means that $\hat{f}(Z_{1:m})$ is not independent of $Z_{1:m}$ and certain information exists. Our next Lemma quantifies this statement and, as far the author knows, is novel. The proof is provided in the supplementary material

**Lemma 2.** *Let $X$ and $Y$ be two random variables such that $X$ is bounded by 1, $\mathbb{E}(X) = 0$ and $\mathbb{E}(Y^2) \le 1$. If $\mathbb{E}[XY] = \beta$ then:*

$$\sqrt{I(X,Y)} \ge \frac{\beta^2}{2\sqrt{2}}.$$

The proof of Lemma 2 is provided in the supplementary material

With Lemmas 1 and 2 at hand, the proof idea is quite straightforward. For every $z \in \{-1/\sqrt{d}, 1/\sqrt{d}\}^d$ we define the following loss function:

$$f(w, z) = \sum_{t=1}^{d} (w(t) - z(t))^2 = \|w - z\|^2. \tag{8}$$

For every distribution $D$, one can show that the minimizer $w^\star = \arg\min L_D(w)$ is provided by $w = \mathbb{E}[z]$. By a standard decomposition we can show:

$$
\begin{aligned}
L_D(w_S) - L_D(w^\star) &= \mathbb{E}\left[\|w_S + (w^\star - z) - w^\star\|^2 - \|w^\star - z\|^2\right] \\
&= \mathbb{E}\left[\|w_S - w^\star\|^2 + 2(w_S - w^\star) \cdot (w^\star - z) + \|w^\star - z\|^2 - \|w^\star - z\|^2\right] \\
&= \mathbb{E}\left[\|w_S - w^\star\|^2\right] + 2\mathbb{E}\left[(w_S - w^\star) \cdot (\mathbb{E}[z] - z)\right] \\
&= \mathbb{E}\left[\|w_S - w^\star\|^2\right] \tag{9}
\end{aligned}
$$

Now, for simplicity of this overview we only consider the case that $\Delta(w_S) = \Omega(1)$ is some non-trivial constant, for example we may assume that $\Delta(w_S) < 1/54$ and let us show that for every coordinate, $t$, the mutual information between $w_S(t)$ and $\sum z_i(t)$ is order of $\Omega(1/m^2)$. Then by standard chain rule, and proper derivations the end result can be obtained.

Indeed, one can observe that if $\Delta(w_S) < 1/54$, and since $w^\star = \mathbb{E}[z] = P$ we have that $\mathbb{E}((\sqrt{d}w_S(t) - P(t))^2) \le 1/54$. In turn, we can use Lemma 1, and show that in expectation over $P$ we also have:

$$\mathbb{E}\left[\left(\sqrt{d}w_S(t) - P\right) \cdot \left(\sum_{i=1}^{m}\left(\sqrt{d}z_i(t) - P\right)\right)\right] = \Omega(1).$$

We now use Lemma 2, and convexity, to lower bound the individual sume of the mutual information, $\sum_{i=1}^{m} I(w_S(t); z_i(t)|P)$. By standard technique we conclude that there exists $P$ for which the mutual information is bounded.

The above outline does not provide a bound that scales with the accuracy of $\Delta(w_S)$, and we need to be more careful in our analysis for the full derivation. The detailed proof is provided in the supplementary material.

**Ackgnoweledgments** The author would like to thank the anonymous reviewers that suggested to incorporate the lower bound for the individual mutual information term. This turned out to simplify some of the proofs. The author would like to thank Shay Moran, and the research was funded in part by the ERC grant (GENERALIZATION, 10139692), as well as an ISF Grant (2188 \ 20). Views and opinions expressed are however those of the author(s) only and do not necessarily reflect those of the European Union or the European Research Council Executive Agency. Neither the European Union nor the granting authority can be held responsible for them. The author is a recipient of a Google research scholar award and would like to acknowledge his thanks.

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
