# A    Proof of Lemma 2

The proof relies on the following two well-known and useful results, Pinsker's inequality and the coupling Lemma. For that, we define the total variation between two distributions $P_1$ and $P_2$

$$\|P_1 - P_2\| = \sup_{E \subseteq \mathcal{X}} \left( \sum_{x \in E} P_1(x) - P_2(x) \right).$$

The first Lemma, relates the total variation distance between two distributions to their KL distance (see for example Duchi [2016]):

**Lemma** (Pinsker's inequality). *Let $P_1$ and $P_2$ be two distributions over a finite domain $\mathcal{X}$ then:*

$$\|P_1 - P_2\| \leq \sqrt{\frac{1}{2} D_{KL}(P_1\|P_2)}. \tag{10}$$

The next Lemma will help us to relate the correlation between two r.v.s and the total variation distance, for a proof we refer to Aldous [1983]:

**Lemma** (Coupling Lemma). *Suppose $P_1$ and $P_2$ are two distributions over a finite domain $\mathcal{X}$. Then for any distribution $D$ over joints random variables $X_1, X_2$ that take value in $\mathcal{X}$ such that $D_{X_1} = P_1$ and $D_{X_2} = P_2$: $\|P_1 - P_2\| \leq D(X_1 \neq X_2)$, further there exists a distribution $D$ as above such that:*

$$\|P_1 - P_2\| = D(X_1 \neq X_2). \tag{11}$$

Let $P$ be the joint distribution over two random variables as in Lemma 2. For every $y$, let $X_y$ be a random variable that is distributed according to $P_{X|y}$, namely $P(X_y = x) = P(X = x|y = Y)$.

By the coupling Lemma, there exists a distribution $D_y$ over the joint random variables $X$ and $X_y$ such that

$$\|P_X - P_{X|y}\| = D_y(X \neq X_y).$$

Now, we consider a distribution, $D$, over random variables $X, X_Y, Y$ as follows: first we randomly pick $Y$ according to $P_Y$, then given $y = Y$, we choose $X, X_y$ according to $D_y$. That means, that $X_y$ is distributed according to $P_{X|y}$, and $X$ is distributed according to $P_X$ for every $y = Y$. In particular, $X$ and $Y$ are independent and:

$$\mathbb{E}_D(X \cdot Y) = \mathbb{E}[X] \mathbb{E}[Y] = 0, \tag{12}$$

and $X_Y$ and $Y$ are distributed according to $P$ and

$$\mathbb{E}_D[X_Y Y] = \beta. \tag{13}$$

Next, we have that for every $y$, as $X$ and $X_y$ are bounded by 1:

$$\mathbb{E}_{D_y}[X^2] + \mathbb{E}_{D_y}[X_y^2] - 2\mathbb{E}_{D_y}[X \cdot X_y] = \mathbb{E}_{D_y}[(X - X_y)^2]$$
$$= D_y(X \neq X_y) \mathbb{E}_{D_y}((X - X_y)^2 | X \neq X_y)$$
$$\leq 4 D_y(X \neq X_y)$$
$$= 4\|D_X - D_{X|y}\|$$
$$\leq 4\sqrt{\frac{1}{2} D_{KL}(D_{X|y}\|D_X)} \qquad \text{Pinsker's inequality}$$

Taking expectation over $y$ on both sides, dividing by 2 and by Jensen's inequality:

$$\mathbb{E}_D[X^2] - \mathbb{E}_D[X \cdot X_Y] \leq 2\mathbb{E}_Y\left[\sqrt{\frac{1}{2} D_{KL}(D_{X|y}\|D_X)}\right]$$
$$\leq \sqrt{2I(X,Y)} \tag{14}$$

Next, we write: $X_Y = \beta Y + \left(\sqrt{\mathbb{E}_D[X_Y^2] - \beta^2}\right) Z_Y$, where $Z_Y = \frac{X_Y - \beta Y}{\sqrt{\mathbb{E}_D[X_Y^2] - \beta^2}}$.

Notice that

$$\mathbb{E}[Z_Y^2] \leq \frac{\mathbb{E}[(X_Y - \beta Y)^2]}{\mathbb{E}[X_Y^2] - \beta^2}$$

$$= \frac{\mathbb{E}[X_Y^2] - 2\beta^2 + \beta^2 \mathbb{E}[Y^2]}{\mathbb{E}[X_Y^2] - \beta^2} \qquad\qquad Eq.\,(13)$$

$$\leq 1. \qquad\qquad\qquad \mathbb{E}[Y^2] \leq 1 \qquad (15)$$

Now from Eq. (12) and Cauchy Schwartz, we obtain:

$$\mathbb{E}_D[X \cdot X_Y] = \mathbb{E}_D\left[\left(\sqrt{\mathbb{E}_D[X_Y^2] - \beta^2}\right) X \cdot Z_Y\right] \qquad\qquad Eq.\,(12)$$

$$\leq \sqrt{\mathbb{E}_D[X_Y^2] - \beta^2}\sqrt{\mathbb{E}_D[X^2]\,\mathbb{E}_D[Z_Y^2]} \qquad\qquad C.S$$

$$\leq \sqrt{\mathbb{E}_D[X^2] - \beta^2}\sqrt{\mathbb{E}_D[X^2]} \qquad\qquad Eq.\,(15). \qquad (16)$$

Plugging the above in Eq. (14) we get (where we supress the subscript $D$ in $\mathbb{E}_D$):

$$\sqrt{I(X,Y)} \geq \frac{1}{\sqrt{2}}\,\mathbb{E}[X_Y^2] - \frac{1}{\sqrt{2}}\,\mathbb{E}[X \cdot X_Y] \qquad\qquad Eq.\,(14)$$

$$\geq \frac{1}{\sqrt{2}}\,\mathbb{E}[X_Y^2] - \frac{1}{\sqrt{2}}\sqrt{\mathbb{E}[X^2]}\sqrt{\mathbb{E}[X^2] - \beta^2} \qquad\qquad Eq.\,(16)$$

$$= \sqrt{\frac{\mathbb{E}[X^2]}{2}}\left(\sqrt{\mathbb{E}[X^2]} - \sqrt{\mathbb{E}[X^2] - \beta^2}\right)$$

$$= \frac{\sqrt{\mathbb{E}[X^2]} + \sqrt{\mathbb{E}[X^2]}}{2\sqrt{2}}\left(\sqrt{\mathbb{E}[X^2]} - \sqrt{\mathbb{E}[X^2] - \beta^2}\right)$$

$$\geq \frac{\sqrt{\mathbb{E}[X^2]} + \sqrt{\mathbb{E}[X^2] - \beta^2}}{2\sqrt{2}}\left(\sqrt{\mathbb{E}[X^2]} - \sqrt{\mathbb{E}[X^2] - \beta^2}\right)$$

$$\geq \frac{\beta^2}{2\sqrt{2}}.$$

# B  Proof of Theorem 1

For a vector $p \in [-1/3, 1/3]^d$ we define a distribution $D(p)$ where $z \sim D(p)$ is such that $z \in \{1/\sqrt{d}, \sqrt{d}\}^d$ and each coordinate $z(t)$ is chosen uniformly such that $\mathbb{E}[z(t)] = p(t)/\sqrt{d}$. The loss function is the same as in Eq. (8), namely:

$$f(w, z) = \|w - z\|^2.$$

Next, given an algorithm $A$ and positive $\varepsilon < 1/54$, choose $m$ such that $m > m(\varepsilon)$. Then, from the output of the algorithm we construct an estimator of $p$, by letting

$$\hat{p}(t) = \begin{cases} \sqrt{d}w_S(t) & |\sqrt{d}w_S(t)| \leq 1 \\ \mathrm{sgn}(\sqrt{d}w_S(t)) & \text{o.w.} \end{cases}.$$

Using the inequality developed in Eq. (9), and because $|p(t)| \leq 1$, we have that

$$\mathbb{E}\left[\|\hat{p}(t) - p(t)\|^2\right] \leq \mathbb{E}\left[\|\sqrt{d}w_S(t) - p(t)\|^2\right] \leq d\,\mathbb{E}\left[L_D(w_S) - L_D(w^\star)\right] \leq d\,\mathbb{E}[\Delta_D(w_S)] \leq d\varepsilon \tag{17}$$

Next, for fixed $p$ we define two random variables:

$$X_p(t) = \frac{1 - 9p(t)^2}{9 - 9p(t)^2} \sqrt{\mathbb{E}\left((\hat{p}(t) - p(t))^2\right)} \sum_{i=1}^{m} (\sqrt{d}z_i(t) - p), \quad Y_p(t) = \frac{1}{\sqrt{\mathbb{E}\left((\hat{p}(t) - p(t))^2\right)}} (\hat{p}(t) - p(t)).$$

$$(18)$$

We now apply Lemma 1 with $Z_i = z_i(t)$, and

$$\hat{f}(Z_{1:m}) = f(z_1(t), \ldots, z_m(t)) := \hat{p}(t) \quad \text{and,} \qquad P = p(t).$$

Notice that $\hat{f}$ is not necessarily a deterministic function of $z_1(t), \ldots, z_m(t)$ as it is allowed to depend on the independent random variables $z_i(j)$ with $j \neq t$. Nevertheless, applying Lemma 1 to each realization of $\hat{f}$ then by the definition of $X_p$ and $Y_p$, we have that for uniformly chosen $p$ and uniformly chosen coordinate $t$, we have that

$$\mathbb{E}_{p,t}\left[\mathbb{E}_{S \sim D^m(p)}\left[X_p(t)Y_p(t)\right]\right] \geq \frac{1}{27} - \mathbb{E}_{p,t}\left[\mathbb{E}_S(\hat{p}(t) - p(t)^2)\right] \geq \frac{1}{27} - \varepsilon \geq \frac{1}{54}, \qquad (19)$$

and we have the following second moment bound, via Cauchy Schwartz:

$$\mathbb{E}_{p,t}\left[\left(\mathbb{E}_{S \sim D^m(p)}[X_p(t)Y_p(t)]\right)^2\right] \leq \mathbb{E}_{p,t}\left[\mathbb{E}_S[X_p^2(t)]\,\mathbb{E}_S[Y_p^2(t)]\right] \leq m\varepsilon. \qquad (20)$$

Write $Z = \mathbb{E}_{S \sim D^m(p)}\left[X_p(t)Y_p(y)\right]$, and Apply Paley-Zygmund inequality [Paley and Zygmund, 1932] (see Remark 1 for the exact version of the inequality we use here), to obtain: :

$$\begin{aligned}
\mathbb{P}_{p,t}\left(\mathbb{E}_{S \sim D^m(p)}\left[X_p(t)Y_p(t)\right] \geq \frac{1}{2} \cdot \frac{1}{54}\right) &\geq \mathbb{P}_{p,t}\left(Z \geq \frac{1}{2} \cdot \mathbb{E}_{p,t}[Z]\right) && Eq.\,(19) \\
&\geq (1 - 1/2)^2 \cdot \frac{\mathbb{E}_{p,t}[Z]^2}{\mathbb{E}_{p,t}[Z^2]} && \text{Paley-Zygmund} \\
&= \frac{1}{4}\frac{1}{54^2 m\varepsilon} && Eqs.\,(19)\,and\,(20) \\
&\geq \frac{1}{10^6 m\varepsilon}. && (21)
\end{aligned}$$

We conclude that for a random pair $(p, t)$, with probability at least $\frac{1}{10^6 m\varepsilon}$, we have that

$$\mathbb{E}[X_p(t)Y_p(t)] \geq \frac{1}{108}.$$

In particular, there exists a vector $p$ such that, by taking expectation over $t$, for $\frac{d}{10^6 m\varepsilon}$ of the coordinates, $t$, we have that

$$\mathbb{E}_S[X_p(t)Y_p(t)] \geq \frac{1}{108}. \qquad (22)$$

Let us fix the above $p$, for the choice of algorithm $A$ and $\varepsilon > 0$, and we assume now that the data is distributed according to $D(p)$ and we denote by $\mathcal{G}(p)$ the set of coordinates $t \in [d]$ that satisfy Eq. (22). In particular we have that

$$|\mathcal{G}(p)| \geq \frac{d}{10^6 m\varepsilon}. \qquad (23)$$

Next, given a sample $z_1, \ldots, z_m$, let us denote $\mathbf{z}_i^{(t)}$ the tuple, $\mathbf{z}_i^{(t)} = (z_i(1), \ldots, z_i(t-1))$. We apply standard chain rule to obtain:

$$
\begin{aligned}
\sum_{i=1}^{m} I(w_S; z_i) &= \sum_{i=1}^{m} \sum_{t=1}^{d} I\left(w_S; z_i(t) \,\Big|\, \mathbf{z}_i^{(t)}\right) \\
&\geq \sum_{i=1}^{m} \sum_{t=1}^{d} I\left(w_S(t); z_i(t) \,\Big|\, \mathbf{z}_i^{(t)}\right) && \text{info processing} \\
&= \sum_{t=1}^{d} \sum_{i=1}^{m} I\left(\left(w_S(t), \mathbf{z}_i^{(t)}\right); z_i(t)\right) - I\left(\mathbf{z}_i^{(t)}; z_i(t)\right) && \text{chain rule} \\
&= \sum_{t=1}^{d} \sum_{i=1}^{m} I\left((w_S(t), \mathbf{z}_i^{(t)}); z_i(t)\right) && \mathbf{z}_i^{(t)} \perp z_i(t) \\
&\geq \sum_{t \in \mathcal{G}(p)} \sum_{i=1}^{m} I(w_S(t); z_i(t)) && \text{info processing} \quad (24)
\end{aligned}
$$

Next, we define:

$$
X_p^i(t) = \frac{1 - 9p(t)^2}{9 - 9p(t)^2} \sqrt{\mathbb{E}\left((\hat{p}(t) - p(t))^2\right)}(\sqrt{d} z_i(t) - p).
$$

Notice that

$$
|X_p^i(t)| \leq 2\sqrt{\mathbb{E}\left((\hat{p}(t) - p(t))^2\right)},
$$

and that $\mathbb{E}\left[Y_p(t)^2\right] \leq 1$. Also, notice, that as $p$ is fixed, $X_p^i(t)$ is determined by $z_i(t)$ and similarly $Y_p(t)$ is determined by $w_S(t)$, we thus have by information processing inequality:

$$
\begin{aligned}
\sum_{i=1}^{m} I(w_S; z_i) &\geq \sum_{i=1}^{m} \sum_{t \in \mathcal{G}(p)} I(w_S(t); z_i(t)) && Eq.\ (24) \\
&\geq \sum_{i=1}^{m} \sum_{t \in \mathcal{G}(p)} I\left(\frac{X_p^i(t)}{2\sqrt{\mathbb{E}\left((\hat{p}(t) - p(t))^2\right)}}; Y_p(t)\right) && \text{data processing inequality} \\
&\geq \sum_{i=1}^{m} \sum_{t \in \mathcal{G}(p)} \frac{\mathbb{E}[X_p^i(t) Y_p(t)]^4}{128 \,\mathbb{E}\left[(\hat{p}(t) - p(t))^2\right]^2} \\
&\geq m \sum_{t \in \mathcal{G}(p)} \frac{\mathbb{E}[\frac{1}{m} \sum_{i=1}^{m} X_p^i(t) Y_p(t)]^4}{128 \,\mathbb{E}\left[(\hat{p}(t) - p(t))^2\right]^2} && \text{convexity} \\
&\geq \sum_{t \in \mathcal{G}(p)} \frac{\mathbb{E}[X_p(t) Y_p(t)]^4}{128 m^3 \,\mathbb{E}\left[(\hat{p}(t) - p(t))^2\right]^2} && X_p(t) = \sum_{i=1}^{m} X_p^i(t)
\end{aligned}
$$

Next we apply Eq. (22) that shows that $\mathbb{E}[X_p(t)Y_p(t)] \geq 128$ and continue the analysis:

$$
\begin{aligned}
\sum_{i=1}^{m} I(w_S; z_i) &\geq \sum_{t \in \mathcal{G}(p)} \frac{\mathbb{E}[X_p(t)Y_p(t)]^4}{128 m^3 \, \mathbb{E}\left[(\hat{p}(t) - p(t))^2\right]^2} \\
&\geq \sum_{t \in \mathcal{G}(p)} \frac{1}{128^5 m^3 \, \mathbb{E}\left[(\hat{p}(t) - p(t))^2\right]^2} &\quad Eq.\ (22) \\
&\geq |\mathcal{G}(p)| \left( \frac{1}{128^5 m^3} \frac{1}{\left( \frac{1}{|\mathcal{G}(p)|} \sum_{t \in \mathcal{G}(p)} \mathbb{E}[(\hat{p}(t) - p(t))^2] \right)^2} \right) &\quad \text{Convexity of } 1/(x)^2, \text{ for } x > 0 \\
&\geq \frac{|\mathcal{G}(p)|^3}{128^5 m^3 \, \mathbb{E}[\|\hat{p}(t) - p(t)\|^2]^2} \\
&\geq \frac{|\mathcal{G}(p)|^3}{128^5 m^3 d^2 \varepsilon^2} \\
&\geq \frac{d}{128^5 \cdot 10^{18} \cdot m^6 \varepsilon^5} &\quad Eq.\ (22),
\end{aligned}
$$

(25)

Overall then, we have that:

$$
\sum_{i=1}^{m} I(w_S; z_i) = \tilde{\Omega}\left( \frac{d}{m^6 \varepsilon^5} \right)
$$

**Remark 1** (Remark on Paley-Zygmund inequality). *Paley Zygmund inequality states that for a random varialbe $Z$:*

$$
\mathbb{P}(Z > \theta \, \mathbb{E}[Z]) \geq (1 - \theta)^2 \frac{\mathbb{E}[Z]^2}{\mathbb{E}[Z^2]}.
$$

*It is often assumed that $Z$ is a non-negative random variable for the inequality to hold. It is not hard to see that the inequality holds for* any *random variable with nonnegative expectation (which is our case here). Indeed, the two-line proof goes as follows:*

*First,*

$$
\mathbb{E}[Z] = \mathbb{E}[Z\mathbf{1}_{Z \leq \theta \, \mathbb{E}[Z]}] + \mathbb{E}[Z\mathbf{1}_{Z > \theta \, \mathbb{E}[Z]}] \leq \theta \, \mathbb{E}[Z] + \mathbb{E}[Z\mathbf{1}_{Z > \theta \, \mathbb{E}[Z]}].
$$

*The second term in RHS is at most $\mathbb{E}[Z^2]^{1/2} \cdot \mathbb{P}(Z > \theta \, \mathbb{E}[Z])^{1/2}$ by Cauchy Schwartz inequality. The desired inequality then follows by by rearranging terms, dividing in $\mathbb{E}[Z^2]^{1/2}$ and taking square of both sides (which requires positivity of $\mathbb{E}[Z]$).*