# OpenReview forum: "Information Theoretic Lower Bounds for Information Theoretic Upper Bounds"
_NeurIPS.cc/2023/Conference — NeurIPS 2023 poster_

### Official Review · Reviewer_dYAB · 2023-07-02

**Soundness:** 2 fair
**Presentation:** 3 good
**Contribution:** 2 fair
**Rating:** 6
**Confidence:** 4

**Summary:**

This paper provides a lower bound on the mutual information between the output weights and the input training data in the context of stochastic convex optimization to examine the tightness of the mutual information-based generalization bound. It is shown in the paper that mutual information grows with the dimension of the parameter, which implies that existing information-theoretic generalization bounds fall short in capturing the generalization capabilities of SGD and regularized ERM, which have dimension-independent sample complexity.

**Strengths:**

It is important to consider the lower bound (converse) result to understand the fundamental limits of information-theoretic analysis in characterizing the generalization of learning algorithms. To my understanding, pessimistic analysis can be more insightful than some ad-hoc method for tightening existing bounds.

**Weaknesses:**

1.	It is hard for me to understand why Theorem 1 holds for any learning algorithm. Let us take the Gibbs algorithm considered in Section 4.3 of (Xu and Raginsky [2017]) and further investigated in the following paper as an example.

Aminian, Gholamali, Yuheng Bu, Laura Toni, Miguel Rodrigues, and Gregory Wornell. "An exact characterization of the generalization error for the Gibbs algorithm." Advances in Neural Information Processing Systems 34 (2021): 8106-8118.

The above reference provides an exact characterization of generalization error using information measures. It has been shown in their proof of Theorem 2 that the mutual information $I(S;W_S)$ can be upper bound by $O(1/n)$, which is independent of d. Thus, it is hard for me to digest the result presented in Theorem 1.

2.	This paper mainly criticizes the tightness of the MI bound in (Xu and Raginsky [2017]) and the CMI bound in (Steinke and Zakynthinou [2020]). However, these issues are well-known in the literature. For example, MI can be infinite for deterministic learning algorithms. There are results tightening these results by considering the MI (or CMI) between W and each individual sample $Z_i$ (see below). Is there a way to show that these bounds also suffered from the same issue described in the paper?

Bu, Yuheng, Shaofeng Zou, and Venugopal V. Veeravalli. "Tightening mutual information-based bounds on generalization error." IEEE Journal on Selected Areas in Information Theory 1, no. 1 (2020): 121-130.

Zhou, Ruida, Chao Tian, and Tie Liu. "Individually conditional individual mutual information bound on generalization error." IEEE Transactions on Information Theory 68, no. 5 (2022): 3304-3316.

**Questions:**

1.	Line 212, to my understanding it should be d^{1/7}/m^{6/7}. Please double-check.
2.	Line 280, it is said that CMI_m(A)=O(m). Can you provide a reference or insights into why it is the case?
3.	Line 20 “inorder” should be “in order”.
4.	The term suboptimality is often referred to as excess risk in literature.

**Limitations:**

No potential negative societal impact was observed.

---

> ### Author Rebuttal · Authors · 2023-08-08
>
> Thank you very much for your detailed review. It seems the main weakness pointed out is an alleged contradiction between the result and existing upper bounds. There is no such contradiction and hopefully below it will be clarified, please do ask for further clarifications if this is not the case or if other issues were found to be emergent and are not resolved.
>
> > It is hard for me to understand why Theorem 1 holds for any learning algorithm. Let us take the Gibbs algorithm considered in Section 4.3 of (Xu and Raginsky [2017]) and further investigated in the following paper as an example.
>
> There is no contradiction between the result in this paper and the analysis of the Gibbs algorithm. Notice that the bound on the information scales with the temperature hyper-parameter (\alpha). In more detail, if the temperature is very low then the Gibbs algorithm behaves like an ERM algorithm and has no generalization guarantees. At the other extreme, if the temperature is very high the algorithm behaves close to purely random and one can achieve a non trivial generalization error/ MI bound, but in this extreme there will be a trivial train error - and as a result, trivial test error..
>
> Notice that theorem 1 applies only to an algorithm with non-trivial **test error**. So there’s no contradiction and the conclusion is that, to obtain non trivial train error, the temperature must scale with the dimension. Notice also that it is trivial to obtain small MI-bounds to algorithms with trivial test error (e.g. random guessing) so the assumption of small test error is in fact necessary here.
>
> > However, these issues are well-known in the literature. For example, MI can be infinite for deterministic learning algorithms. There are results tightening these results by considering the MI (or CMI) between W and each individual sample  Z_i (see below). Is there a way to show that these bounds also suffered from the same issue described in the paper?
>
> Thanks for the reference. First the issue shown in this paper is not that there are algorithms that carry information but that it is necessary to carry information for learning. The paper does not rule out the possibility of exploiting MI-type bounds in general, but to simply observe that some excess information is necessary.
>
> Regarding the specific variation proposed in Bu et al. Thanks for the reference. The argument depicted here can be easily applied to their setup. As this was asked by several reviewers I will provide a proof sketch in a separate official comment (a full derivation can also be provided upon request)
>
>
> >It is said that CMI_m(A)=O(m). Can you provide a reference or insights into why it is the case?
>
> .Given Z the entropy of S is merely the chosen example (i.e. z_i^1 or z_i^0), this is a r.v.with  2^m possible configurations hence entropy at most m.

---

> > ### Comment · Reviewer_dYAB · 2023-08-21
> > **Thanks for the response**
> >
> > I thank the authors for providing a detailed response. The rebuttal addresses all my concerns, and I would like to increase my score to 6.
> >
> > Although the review process does not allow the reviewer to enforce any required changes, I would like to emphasize that the authors should include the discussion on the Gibbs algorithm and the result of individual samples bound into the final version of the main paper. Also, it would be insightful to add a concluding section to discuss the interpretation of this lower-bound result and possible future directions. I feel that this kind of converse analysis can be very helpful in improving the existing understanding of generalization, and I
> > echo with the author, "The paper does not rule out the possibility of exploiting MI-type bounds in general, but to simply observe that some excess information is necessary for some distributions."

---

> > > ### Author Response · Authors · 2023-08-21
> > > **Thanks**
> > >
> > > Thank you very much,
> > >
> > > A discussion on the Gibbs algorithm will be further expanded as well as the result of individual samples, Other advices suggested by the reviewer (as well as the rest of the reviewers) will be followed.

---

### Official Review · Reviewer_UMeF · 2023-07-03

**Soundness:** 3 good
**Presentation:** 2 fair
**Contribution:** 3 good
**Rating:** 6
**Confidence:** 3

**Summary:**

This paper challenges the tightness of some information-theoretic upper bounds on the generalization error in the stochastic convex optimization (SCO) setting (i.e. convex, Lipschitz, bounded loss on a bounded domain). More precisely, it challenges the bound from Xu and Raginksy that (vaguely) states that the generalization error is bounded from above by a function of the mutual information between the dataset and the output of the learning algorithm. With this purpose, the authors derive lower bounds on the mutual information that grows with the dimension of the input space. This, in turn, shows that there is a distribution over an input space such that the mutual information is arbitrarily large. Therefore the mutual information-based bounds are vacuous for that distribution.

These lower bounds are obtained by means of three essential tricks:
1) Noting that one may only study discrete algorithms and discrete distributions over the inputs. The fact of considering discrete distributions over the input comes naturally from the fact that they are looking for lower bounds. Then, restricting to discrete algorithms is possible since (i) the loss is Lipschitz and the generalization error of a discretization of the parameters can at most incur into a constant multiplicative penalty and since (ii) the mutual information will only be smaller by the data processing inequality.
2) Employing a fingerprinting lemma from Kamath et al. (2019). For random variables $(X\_i)\_{i=1}^m \in \lbrace -1, 1 \rbrace$ such that $\mathbb{E}[X\_i]=p$. This lemma essentially states that the MSE of an estimator of $p$ decreases the more correlated the error of the estimator is with the error of the empirical mean. That is, a good estimator of $p$ is highly correlated with the empirical mean.
3) Noting that if two random variables $X$ and $Y$ are correlated with $\mathbb{E}[XY] = \beta$, then the mutual information $I(X;Y)$ is grows as $\Omega(\beta^4)$.

Overly simplifying, this way, they construct an example where the loss function is $f(w,z) = \lVert w - z \rVert^2$. Then, by the fingerprinting lemma, any algorithm that attains a good error is necessarily correlated with the empirical mean. Therefore, the mutual information between the output of the algorithm and the empirical mean is bounded from below. Finally, by the data processing inequality, the mutual information between the output of the algorithm and the dataset is also bounded from below. The use of the chain rule makes sure that this scales with the dimension of the input.

**Strengths:**

The problem studied is interesting. While Bassily et al. (2019) presented some lower bounds of the mutual information for learning thresholds and Haghifam et al. (2022) showed lower bounds of the mutual information, conditional mutual information, and its variants for gradient descent (GD) in the SCO setting, studying general lower bounds for any algorithms in this setting is important.

As mentioned in the summary, the realization of being able to only consider discrete algorithms and the usage of the fingerprinting lemma to find lower bounds on the mutual information of algorithms in the SCO setting is original and interesting. Moreover, formalizing the expected result that correlated random variables have large mutual information is valuable in its own right.

**Weaknesses:**

- The authors sometimes fail to position the related literature correctly.
  - They only glass over Haghifam et al. (2022) in Section 1.1 and they say that the paper finds lower bounds on the mutual information (MI) and the conditional mutual information (CMI) of GD and a variant of GD with a perturbation on the final iterate. They also find lower bounds for the individual sample versions of the MI and CMI as well as the evaluated CMI for the GD algorithm.  Moreover, they mention that the bounds have a logarithmic dependence on the dimension. This is not accurate, while the dependence is logarithmic for the perturbed version of GD, for vanilla GD the bounds grow with the square root of the dimension.
  - In line 257 it seems they forgot to cite Bassiliy et al. (2018). In the same sub-subsection (CMI-bounds) it seems that they also forgot to mention the existing lower bounds from Haghifam et al. (2022).
  - Also in the CMI-bounds sub-subsection they describe wrong the CMI. The CMI is not $I(w\_S;S|Z)$, it is $I(w\_S; U|Z)$, where $U$ is a sequence of Bernoulli random variables whose realization is used to select between $z\_i^0$ and $z\_i^1$ to generate the dataset $S$.

- The paper is not very clear.
  - Most of the citations in the paper are not in the correct format (that is, they are not within brackets in the \citep style of natbib, but directly written in the \citet style of natbib), difficulting readability.
  - Sometimes the sentences are a little overstated: e.g. in line 174 they say "Theorem 1 shows that any algorithm with non-trivial learning guarantees must carry a dimension-dependent amount of information on the sample or require a large sample". This is not completely accurate. To be precise, for every algorithm $A$, there is a distribution $D$ over a space $\mathcal{Z}$ and a convex, Lipschitz function $f$ such that if the algorithm achieves a non-trivial learning guarantee, then it must carry a dimension-dependent amount of information on the sample or require a large sample.
  - The sub-subsection CMI-bounds seems a little disconnected. The whole paper focuses on finding lower bounds of the mutual information, showing limitations to these bounds. Haghifam et al. (2022) show that at least for GD one can have a population risk of $O(1/\sqrt{m})$ while having a CMI of $\Omega(m)$, which would go in the direction of obtaining lower bounds. Moreover, it is not clear why $O(1/\sqrt{m})$ is chosen for the population risk and $o(\sqrt{m})$ is chosen for the CMI. The subsampling statement above seems to try to clarify this, but why would not one choose to subsample $O(m^{1/4})$ or $O(\log m)$? There seems to be no justification.
  - Lemma 3 is used without any proof in the Appendix of the supplementary material, which is in the template of the submission to another conference whose main text differs in some places from the main text submitted.
  - The final part of the paper (from line 333 on) seems disconnected from the rest of the text.

Small mistakes:
-  In 237-238 it states that "for any algorithm such that $\Delta\_S(w\_S) = O(1/m)$ we will have that the bound in Eq. (2) is order of $\mathbb{E}[\Delta\_D (w\_S)] = \tilde{O}(\sqrt{d/m})$. This is not correct, the bound is of that order, but not $\mathbb{E}[\Delta\_D (w\_S)] $.
- In Appendix A, it should be $\lbrace -1 / \sqrt{d}, 1 / \sqrt{d} \rbrace$, right?


**Questions:**

- Why is there a vector $p$ such that for $d / (10^6 m \epsilon)$ of the coordinates (22) holds? Why is it not possible that less cordinates have that value be much larger than 1/108?
- Why can you employ the data processing inequality after (24)? X\_p(t) depends both on $Z\_i$ and $W\_S$, and $Y\_p(t)$ depends on $W\_S$?
- Could the techniques from this paper be employed to also find lower bounds on stronger information-theoretic bounds on the generalization such as the individual MI from Bu, Zhou, and Veeravalli (2020)?

**References**

Y.Bu, S. Zhou, and V. V. Veeravalli. "Tightening Mutual Information-Based Bounds on Generalization Error". Journal on Selected Areas in Information Theory. 2020.

**Limitations:**

Some limitations are touched on in the comparison of the results with uniform convergence bounds in lines 239-256. However, there is not a large discussion of the limitations of the paper. For instance, what happens if an algorithm chooses another loss function? or, could one just consider a stronger information-theoretic notion such as the individual MI and then achieve bounds with a good rate?

Maybe the space from lines 333 onwards could be used to reflect more on these issues in the Discussion section.

---

> ### Author Rebuttal · Authors · 2023-08-08
>
>
> Thanks for the review. Hopefully, the rebuttal will help the reviewer be convinced that there is no reason for providing such a low score. Especially given that the reviewer identifies strengths (important problem and interesting techniques) and that the weaknesses can be easily addressed and require some small formatting changes, adding further elaborations, as well as a few citations.
>
> Below is an address to what seem like the most emergent issues, further elaboration can be provided if needed.
>
> > They only glass over Haghifam et al. (2022) in Section 1.1
>
> The paragraph will be changed to clarify that they can achieve bounds to other variants. The final sentence will be clarified: that the logarithmic bound applies to the perturbed version.
>
> > In line 257 it seems they forgot to cite Bassiliy et al. (2018) …. it seems that they also forgot to mention the existing lower bounds from Haghifam et al. (2022).
>
> A discussion on the work of Bassiley et al, in the context of CMI,  will be added here, and also the work of Haghifam et al will be reiterated.
>
> > Also in the CMI-bounds sub-subsection they describe wrong the CMI.
>
> Thanks for the clarification, this detail can be fixed. This is an inaccuracy in the discussion part, it is self-contained and has little to no effect on the rest of the paper (which does not deal with the CMI bound)
>
> > Sometimes the sentences are a little overstated: e.g. in line 174 they say "Theorem 1 shows that any algorithm with non-trivial learning guarantees must carry a dimension-dependent amount of information on the sample or require a large sample".
>
> This sentence appears in the discussion section, after stating the exact statement, and after establishing the paper's distribution-independent setup (a standard arrangement) in previous sections. In this context, which is distribution independent, there's no overstatement. The sentence, though, can be rephrased as: "any algorithm with distribution-independent nontrivial guarantees..."
>
> >Lemma 3 is used without any proof in the Appendix of the supplementary material, which is in the template of the submission to another conference whose main text differs in some places from the main text submitted.
>
> Thanks for this input, there was indeed a mistake in the main text version that was attached to the appendix, apologies for the inconvenience and confusion. This will be fixed ofcourse, and the proof of Lemma 3 will be added to the appendix. For now, the proof does appear in the current reviewed manuscript exactly under the statement of Lemma 3 in the supplementary.
>
>
>
> Questions:
>
> >Why is there a vector p such that for  (d/106m\epsilon^2)  of the coordinates (22) holds? Why is it not possible that less cordinates have that value be much larger than 1/108?
>
> Notice that in (22) we take expectation only over the sample, i.e. one should think of (22) as an event that depends on the realization of (p,t). Thus, if for every p, (22) holds for less than (d/106m\epsilon^2) of coordinates, then for every p, the probability of (22) to hold, conditioned on p and w.r.t t,  is less than (1/106m\epsilon^2). Then, the probability (w.r.t p and t) of (22) to hold is less than (1/106m\epsilon^2) which is a contradiction.
>
> >Why can you employ the data processing inequality after (24)?
>
> Notice that X_p(t) doesn’t depend on w_s as you write. (It depends on the variance of w_S but that is not a random variable but a fixed number determined by p and t)
>
> >Could the techniques from this paper be employed to also find lower bounds on stronger information-theoretic bounds
>
> Yes, thank you very much for this question and for pointing out the bound of Bu et al. Because this was inquired by several reviewers, I will add an additional official comment that discusses this (and a full derivation can be added upon request).

---

> > ### Comment · Reviewer_UMeF · 2023-08-14
> > **Answer to rebuttal**
> >
> > Thank you for your rebuttal.
> >
> > I agree with the authors that the score given was lower than needed. My main motivation for the score was the lack of a proof of Lemma 3, which in turn was needed to prove Lemma 2, which was needed to prove Theorem 1 (the main result of the paper).  Admittedly, having the supplementary material with a different formatting, and a main text very similar to the main text submitted but with some distinctions confused me. It made me look only at the Appendix of the pdf given as the supplementary material. Now I realized that the proof of Lemma 3 **is given** in the main text of the supplementary material, something I missed in the first round. The proof seems correct. This essentially dissipates my main concern. I apologize for missing that and I will take that into account for the final evaluation.
> >
> > Most of my minor issues and questions are also addressed.
> >
> > Some smaller things:
> > * My question and doubts about the CMI subsection remain. Could you discuss them a little further?
> > * I still believe the statement in 174 is overstated. The statement "any algorithm with non-trivial learning guarantees must carry a dimension-dependent amount of information on the sample or require a large sample" seems to imply that this always happens. But it is only shown that there is at least a situation where it happens. To me something of the style "for any algorithm with non-trivial learning guarantees, there is at least one scenario where it must carry a dimension-dependent amount of information on the sample or require a large sample" would represent better the presented result. Do you agree?

---

> > > ### Author Response · Authors · 2023-08-16
> > > **Thanks**
> > >
> > > Thanks, and again apologies for the confusions in the uploaded manuscript.
> > >
> > > > My question and doubts about the CMI subsection remain. Could you discuss them a little further?
> > >
> > > A negative answer to the open problem will show that any sample-efficient learning algorithm must carry at least $\Omega(\sqrt{m})$ CMI and in turn a non-tight CMI generalization error bound of $O(1/\sqrt[4]{m})$. A $\Omega(\sqrt{m})$ lower bound will then be analogue to the $\Omega(d)$ limitation of the MI bound.
> > >
> > > The discussion above the open problem shows that, unlike for MI, $\Omega(\sqrt{m})$ is not necessary for non-trivial population loss (by subsampling for example $\log m$ examples): But that is at the expense of the training error -- which is also a bound on the population error. The author conjecture that the training error and CMI bound will always be larger than $\Omega(1/\sqrt[4]{m})$. A weaker conjecture (which is the focus of the open problem and captures the important cases) is that just for the optimal learning algorithms, the above bound (training error + CMI) will be larger than $\Omega(1/\sqrt[4]{m})$.
> > >
> > > Granted, there are other open problems to be resolved here. For example, as suggested by the reviewer, to investigate the feasible tuples (training error, CMI bounds), and whether subsampling provides the pareto-optimal frontier.
> > >
> > > >  "for any algorithm with non-trivial learning guarantees, there is at least one scenario where it must carry a dimension-dependent amount of information on the sample or require a large sample"
> > >
> > > The sentence can be rephrased as such.

---

> > > > ### Comment · Reviewer_UMeF · 2023-08-18
> > > > **Score update**
> > > >
> > > > Dear authors,
> > > >
> > > > The scores can now be updated. Since my major concern (no proof of Lemma 3) is taken care of, and most of my minor concerns are also addressed, I improved it by 2 points.

---

> > > > > ### Author Response · Authors · 2023-08-20
> > > > > **Thank you!**
> > > > >
> > > > > Thanks!

---

### Official Review · Reviewer_z3UU · 2023-07-05

**Soundness:** 2 fair
**Presentation:** 2 fair
**Contribution:** 3 good
**Rating:** 5
**Confidence:** 3

**Summary:**

This work considers the setting of stochastic convex optimization in $\mathbb{R}^d$ with learning algorithms that achieve less than $\epsilon$ expected excess risk when at least $m(\epsilon)$ examples are given. The main result of this work (Theorem 1) states that such algorithms have $\tilde{\Omega}\left(\frac{d}{\epsilon^5 m(\epsilon)^5}\right)$ amount of Shannon mutual information between the input and the output (i.e., $I(w_S; S)$ with $w_S$ being the output on input sample $S$). The significance of this lower bound is that there is a linear dependence on dimension $d$, which renders the bound of Xu and Raginsky [2017] dimension-dependent in case $m(\epsilon)=o(d^{1/5})$. For example, if one considers minimax optimal learning algorithms (that have $\frac{1}{\epsilon}$ or $\frac{1}{\epsilon^2}$ sample complexity depending on assumptions about the loss function), the generalization gap bounds with input-output mutual information become dimension dependent and can be vacuous when $d \gg m$.

**Strengths:**

**Strength #1: Significance.** Recently information-theoretic generalization bounds have gained a lot of attention, partly because they are algorithm and distribution-dependent and some variants of them are nonvacuous in practical setting for deep learning. Understanding limitations of information-theoretic generalization bounds is thus important and relevant for the NeurIPS community. The lower bound derived in this work is a good contribution to the list of already known limitations.

**Strength #2: Originality.** To my best knowledge the derived lower bound is novel. The related work is cited adequately.

**Weaknesses:**

**Weakness #1: Soundness.** I couldn't verify some parts of the proof of Theorem 1. Some clarifications are needed.
- In the line after equation (17), "Next, for fixed $p$ we define two random variables": Is $p$ a vector here or a constant? Should it be $\frac{1-9p(t)^2}{9-9p(t)^2}$ instead?
- In the line after equation (18), "Applying Lemma 1, ...": It was hard for me to follow the proof. I suggest to clearly write what $f$ is, verify that $f$ has the signature $f: \\{-1,1\\}^m \rightarrow [-1/3, +1/3]$ as required by Lemma 1, and clearly write what $X_i$ of Lemma 1 are in your construction. As I understand $X_i$ of Lemma 1 correspond to $\sqrt{d} z_i(t)$, but in that case $\hat{p}(t)$ is not a function of $\\{\sqrt{d}z_i(t)\\}_{i=1}^m$.
- In the line after equation (20), "Write $Z = \mathbb{E}_{S\sim D^m(p)}\left[ X_p(t) Y_p(t)\right]$": I suggest to use another letter instead of $Z$, which is already used to denote the input. Additionally, in order to apply Paley-Zygmund inequality, $Z$ needs to be positive. I think this should be verified. As I understand, the algorithm can be such that it fails to be positively correlated with the mean for some choices of $p$. For those choices of $p$, $Z$ can be zero or negative.
- I could verify the proof after equation (21).

**Weakness #2: Clarity & Presentation [minor].** The paper would benefit a lot from improvements in terms of clarity and presentation. It was hard for me to read and understand the details, as there were many small mistakes, some parts of derivations were not detailed enough, and some parts of the main text made sense only after reading the later parts of the paper. Please see more detailed comments and suggestions below.

**Weakness #3: Limitations [minor].** Deriving a lower-bound for the input-output mutual information is does not have strong implications about the line of information-theoretic generalization bounds.

It is well-known that information-theoretic generalization bounds that depend on input-output mutual information have many limitations. For example, they can be infinite for well-generalizing continuous learning algorithms. For this reason many techniques of improving such bounds were proposed.
1. Bu et al. [1] derived an improved expected generalization gap bound that depends on $\sum_i I(w_S; Z_i)$ rather than $I(w_S, S)$ with $S=(Z_1,\ldots,Z_m)$ being the sample.
2. As discussed in the paper, there are conditional mutual information bounds that are dimension independent. These CMI bounds of Steinke and Zakynthinou have been improved a lot subsequently: (a) deriving sample wise bounds $\sum_i I(w_S; J_i | \tilde{Z})$ (where $J$ is the train-test split variable and $\tilde{Z} \in \mathcal{Z}^{n\times 2}$ is the supersample) [2], (b) conditioning on individual pair of supersample examples: $\sum_i I(w_S; J_i | \tilde{Z}_i)$ [3]; (c) measuring information in function space [4]; and (d) leave-one-out bounds [5].
To my understanding, the result of this paper does not extend to these stronger bounds.


Final note: the main determinant of the score assigned below is the weakness #1. I am willing to increase the score if the concerns are properly addressed.


**References**

[1] Y. Bu, S. Zou, and V. V. Veeravalli. Tightening mutual information-based bounds on generalization error. IEEE Journal on Selected Areas in Information Theory, 2020.

[2] M. Haghifam, J. Negrea, A. Khisti, D. M. Roy, and G. K. Dziugaite. Sharpened generalization bounds based on conditional mutual information and an application to noisy, iterative algorithms. NeurIPS 2020.

[3] H. Harutyunyan, M. Raginsky, GV. Steeg, and A. Galstyan. Information-theoretic generalization bounds for black-box learning algorithms. NeurIPS 2021.

[4] B. Rodríguez-Gálvez, G. Bassi, R. Thobaben, and M. Skoglund. On random subset generalization error bounds and the stochastic gradient Langevin dynamics algorithm. IEEE ITW 2021.

[5] M. Haghifam, S. Moran, DM. Roy, and GK. Dziugiate. Understanding generalization via leave-one-out conditional mutual information. IEEE ISIT 2022.

**Questions:**

- Please use `\citep` or `\citet` (depending on whether authors are a part of the sentence or not) rather than `\cite`.
- Lines 74-78: Need to be rewritten to improve clarity.
- Line 116: Calling $\Delta_S(w)$ "empirical risk" is confusing. It is better to call it "excess empirical risk".
- Line 116: "With the optimality" -> "with the suboptimality".
- Line 117: Typo in "Leranbility".
- Lines 117-118: As I understand from the expectation in between lines 121-122, only deterministic algorithms are considered. Does the main result of Theorem 1 extend to stochastic algorithms?
- Just below line 187: it should be $\le$ in the first equation line.
- Appendix A. Proof of Theorem 1, first paragraph: It should be $z \in \\{-1/\sqrt{d}, 1/\sqrt{d}\\}$. Also, in "each coordinate z(t) is chosen uniformly" the word "uniformly" should be removed or replaced by "independently".
- Lines 200-201: It is clear what is the main message here, but the information-theoretic bound applies to the generalization gap and should not be directly compared with the suboptimality rate.
- In equation between lines 212 and 213, it should be $d^{1/7}$.
- The CMI bound in eq. (7) instead of $I(w_S; S | Z)$ it should be $I(w_S; J | Z)$ where $J$ denotes the selection variables. Note that $I(w_S; J | Z)  = I(w_S; S | Z) + I(w_S; J | Z, S)$ and the last term is not always zero (e.g., when $\mathcal{Z}$ is a finite domain with small size).
- Line 282: It should be $O(\sqrt{f(m)/m})$.
- Lines 280-286: Subsampling will result in dimension-independent *generalization gap* bounds, but one might not be able to convert that into an excess risk bound.
- In a few places throughout the paper we see "true risk", does it refer to the excess risk (suboptimality) or the expected loss?
- In the equation below line 315, there should be parentheses for the logarithm to indicate that the $\frac{c^2}{\beta^2}$ part is inside the logarithm.

**Limitations:**

See weakness #3.

---

> ### Author Rebuttal · Authors · 2023-08-08
>
> Thanks for your feedback and comments. The proofs are correct, but certain typos have been identified by the reviewer, and indeed, a certain assertion was neglected that could clarify things (see below).If further clarifications are in order, those can be delivered..
>
> > In the line after equation (17), "Next, for fixed p   we define two random variables": Is  a vector here or a constant?
>
> Thanks!, you are right and it is a typo. It should be p(t) instead of p.
>
> > In the line after equation (18), "Applying Lemma 1, ...": It was hard for me to follow the proof. I suggest to clearly write what
>
> Thanks for pointing out the need for further elaboration here. What is missing is the statement that without loss of generality we can assume that $|w(t)| <= 1/(3\sqrt{d})$. Indeed, concatenating w(t) will only diminish the loss and will not add information.This will also mean that f is bounded as required.
>
> Next, as you correctly point out, to apply fingerprinting, f is $\hat{p}(t)$. f is a function of $z_i(t)$’s as $w_S$ is a function of $z_i(t)$, all other r.v.s are independent of $p(t)$ hence the fingerprinting Lemma can be readily applied. This part indeed can benefit from a more detailed derivation but it is well justified
>
>
> > in order to apply Paley-Zygmund inequality,  Z  needs to be positive. I think this should be verified.
>
> Thanks for this remark. Paley-Zygmund is indeed generally formulated so that $Z$ has to be positive. However, it is true that one can apply the inequality to any random variable with positive expectation (i.e. $E[Z]>0$) which follows from  eq. 19.
>
> The proof is so straightforward that it can be verified here: Indeed, the standard derivation begins with the equality:
> $Z = Z\cdot 1_{Z\le \theta E[Z]} + Z\cdot 1_{Z> \theta [Z]}$, hence $E[Z] \le \theta E[Z] + E[ Z\cdot 1_{Z> \theta [Z]}]$.
>
> This step ofcourse doesn’t require positivity. Next, we apply C.S which also doesn’t require positivity:
>  $E[ Z\cdot 1_{Z> \theta [Z]}] \le \sqrt{E[Z^2] P(Z> \theta E[Z])}.$
>
> Rearranging we obtain
>
> $$(1-\theta) E[Z]) \le  \sqrt{E[Z^2] P(Z> \theta E[Z])}.$$
>
> Now taking the square of both sides (which now requires $(1-\theta) E[Z]\ge 0$) we obtain the inequality.
>
> This remark will be added to the manuscript, to make it clear for everyone that Paley-Zygmund can be applied here.
>
> > I could verify the proof after equation (21).
>
> While there are a few tedious lines after eq.21 they are all being justified. Perhaps if the reviewer can point out the difficulty here, that could be gladly clarified.
>
> > It is well-known that information-theoretic generalization bounds that depend on input-output mutual information have many limitations. For example, they can be infinite for well-generalizing continuous learning algorithms.
>
> This is true, but this deficiency is not due to the input-output mutual information, one can artificially encode any information in the model. For example, think of a perfectly well-generalizing algorithm where the model includes storage of all the data that was used in training, such an algorithm will hold information even on the individual samples. The point in this paper is to develop a technique that shows that **every** algorithm must hold information.
>
>  > Bu et al. [1] derived an improved expected generalization gap bound
>
> First, thanks for the reference! The bound in Bu et al indeed requires discussion and that will be added. Similarly, a dimension dependent lower bound can be derived to the form of Bu et al. Since this was asked by several reviewers, a proof sketch will be added in a separate official comment. A full derivation can also be provided upon request.
>
> Regarding further questions, the reviews does identify certain typos and places where clarification is in order, these will be addressed in the paper and Thanks for pointing these out. if some further clarifications are in order, those will gladly be provided.

---

> > ### Comment · Reviewer_z3UU · 2023-08-20
> > **Reviewer response**
> >
> > Thank you for the clarifications. I recommend to improve the proof of Theorem 1 in future revisions, especially where Lemma 1 is applied. It should be mentioned that $f$ is a function of not only $z_i(t)$, but also of $z_i(t')$ where $t'\neq t$. Therefore, one has to first fix these other variables, verify that Lemma 1 applies, apply Lemma 1, and then take expectation over $z_i(t')$ and $t$.
> >
> > With these clarifications and corrections in place, this submission is improved and I will adjust my score accordingly.

---

> > > ### Author Response · Authors · 2023-08-21
> > > **Thanks**
> > >
> > > Thank you very much.
> > >
> > > A revised version will include further elaborations and clarifications following the reviewer's advice.

---

### Official Review · Reviewer_VAde · 2023-07-06

**Soundness:** 3 good
**Presentation:** 3 good
**Contribution:** 4 excellent
**Rating:** 7
**Confidence:** 3

**Summary:**

The paper shows that there exist stochastic convex optimization problems, which are easy to learn but for which the mutual information based generalization bound of [Xu-Raginsky, 2017] scales at least linearly in the dimension of the parameter space.

**Strengths:**

The paper presents the limitation of mutual information based generalization bound of [Xu-Raginsky 2017] in capturing dimension dependency. The work improves on the previous results of [Haghifam et al 2022] in this respect, in which the lower bound scales logarithmically with dimension and is restricted to gradient descent.

I did not carefully check all proofs but they look fine.The results of this paper are original. This work is among the very few that point to the limitations of information-theoretic upper bounds for generalization errors of learning algorithms. Such results are important for the learning theory community to developed improved information-theoretic tools that overcome such limitations.

The paper is also very well written. The presentation is clear to this reviewer. The presented results are very well positioned in the related literature.

**Weaknesses:**

NONE

**Questions:**

NONE

---

> ### Author Rebuttal · Authors · 2023-08-08
>
> Thank you very much!

---

### Author Rebuttal · Authors · 2023-08-08

Thank you very much for the thoughtful reviews.

Several reviewers pointed out to the work of Bu et al. and asked it the techniques can also be applied to this individual sample bound, here I am providing a proof sketch explaining why the technique easily applies. I will only show dependence on the dimension and not on the accuracy (similar to the technical overview in the main text) to avoid complications. Of course, a full derivation as is done in the paper can also be provided. But the argument here already shows that measuring the mutual information with the individual samples will not lead to dimension-free bounds (and that the proof is almost identical).


The main observation is that applying Lemma 2 we have that:

$$ \sum_{i=1}^m \sqrt{I(w_S(t),z_i(t))} = \sum_{i=1}^m \sqrt{I(\sqrt{d} w_S(t)-P(t),\sqrt{d} z_i(t)-P(t))} \ge \sum_{i=1}^m \Omega (E[(\sqrt{d} w_S(t)-P(t)) \cdot (\sqrt{d} z_i(t)-P(t))]^2)$$
and by convexity:
$$ \sum_{i=1}^m \Omega (E[(\sqrt{d} w_S(t)-P(t)) \cdot (\sqrt{d} z_i(t)-P(t))]^2) \ge m \Omega (E[(\sqrt{d} w_S(t)-P(t)) \cdot (\frac{1}{m}\sum_{i=1}^m  \sqrt{d} z_i(t)-P(t))]^2)$$

From here the derivation is the same where we use the fingerprinting Lemma to show that

$$ E[(\sqrt{d} w_S(t)-P(t)) \cdot (\frac{1}{m}\sum_{i=1}^m  \sqrt{d} z_i(t)-P(t))] = \Omega (1) $$ and we conclude that

$$ \frac{1}{m}\sum_{i=1}^m \sqrt{I(w_S(t),z_i(t))}  = \Omega (1) $$
for each single coordinate. By proper derivation we can show that the information accumulated in all coordinates scale with the dimension.

---

### Decision · Program_Chairs · 2023-09-21

**Decision:**

Accept (poster)

**Comment:**

All reviewers agree to accept this paper. The paper is theoretically sound and sufficiently novel to grant acceptance to this conference.

For a camera-ready version, please take into the account the comments from all reviewers regarding typos and clarifications. In particular, please include:
- a discussion about additional results related to Bu et al.
- a proper discussion of Haghifam et al.
- some discussion about Gibbs algorithm